

# Current and future scenarios of suitability and expansion of cassava brown streak disease, *Bemisia tabaci* species complex, and cassava planting in Africa

Geofrey Sikazwe[1,2,3], Rosita Endah epse Yocgo[1,4], Pietro Landi[2,5], David M. Richardson[6,7] and Cang Hui[2,5,8]

[1] African Institute for Mathematical Sciences, Kigali, Rwanda
[2] Department of Mathematical Sciences, University of Stellenbosch, Stellenbosch, South Africa
[3] Mkwawa University College of Education, Iringa, Tanzania
[4] Institute for Plant Biotechnology, Stellenbosch University, Stellenbosch, South Africa
[5] National Institute for Theoretical and Computational Sciences, Stellenbosch University, Stellenbosch, South Africa
[6] Institute of Botany, Czech Academy of Sciences, Průhonice, Czech Republic
[7] Centre for Invasion Biology, Department of Botany and Zoology, Stellenbosch University, Stellenbosch, South Africa
[8] Mathematical Bioscience Unit, African Institute for Mathematical Sciences, Cape Town, South Africa

Corresponding author
Geofrey Sikazwe, geofrey@aims.ac.za

## ABSTRACT

Cassava (*Manihot esculenta*) is among the most important staple crops globally, with an imperative role in supporting the Sustainable Development Goal of 'Zero hunger'. In sub-Saharan Africa, it is cultivated mainly by millions of subsistence farmers who depend directly on it for their socio-economic welfare. However, its yield in some regions has been threatened by several diseases, especially the cassava brown streak disease (CBSD). Changes in climatic conditions enhance the risk of the disease spreading to other planting regions. Here, we characterise the current and future distribution of cassava, CBSD and whitefly *Bemisia tabaci* species complex in Africa, using an ensemble of four species distribution models (SDMs): boosted regression trees, maximum entropy, generalised additive model, and multivariate adaptive regression splines, together with 28 environmental covariates. We collected 1,422 and 1,169 occurrence records for cassava and *Bemisia tabaci* species complex from the Global Biodiversity Information Facility and 750 CBSD occurrence records from published literature and systematic surveys in East Africa. Our results identified isothermality as having the highest contribution to the current distribution of cassava, while elevation was the top predictor of the current distribution of *Bemisia tabaci* species complex. Cassava harvested area and precipitation of the driest month contributed the most to explain the current distribution of CBSD outbreaks. The geographic distributions of these target species are also expected to shift under climate projection scenarios for two mid-century periods (2041–2060 and 2061–2080). Our results indicate that major cassava producers, like Cameron, Ivory Coast, Ghana, and Nigeria, are at greater risk of invasion of CBSD. These results highlight the need for firmer agricultural management

and climate-change mitigation actions in Africa to combat new outbreaks and to contain the spread of CBSD.

## INTRODUCTION

*Manihot esculenta* Crantz (Cassava) from Latin America (Brazil) was introduced to sub-Saharan Africa by Portuguese traders and has increasingly become a major source of calories for over 800 million people worldwide (*FAO, 2013*). Africa contributes at least 56% of the World's total cassava production (*FAO, 2021*), and cassava is the second most important staple food crop in sub-Saharan Africa (*Tomlinson et al., 2018*). Cassava's popularity emanates from its adaptability and capacity to provide acceptable yields under marginal farming conditions and limited water availability. However, various abiotic stressors (temperature and rainfall) and biotic stressors (pests and pathogens) can greatly affect the plant's physiology and growth (*Jones & Barbetti, 2012*) and consequently threaten the Sustainable Development Goal of 'Zero hunger' and socio-economic benefits to millions of resource-limited farmers across Africa. It is therefore imperative to assess the current and future suitability of cassava and its biotic stressors under a changing climate.

Cassava exhibits an array of responses to different climate variables. Temperature, for example, affects various stages of cassava's growth and development, including sprouting, leaf formation, leaf size, and storage root formation. The optimum growth of each stage is reached at an optimum temperature range of 25–35 °C (*El-Sharkawy, 2004*). Sprouting is faster at a soil temperature of about 28–30 °C. However, sprouting is arrested at temperatures above 37 °C and below 17 °C (*El-Sharkawy, 2004*). This partly explains its current cultivation niches, which are predominantly distributed across most tropical and sub-tropical regions. Nevertheless, cassava is a resilient crop and can be cultivated under variable rain-fed conditions, where annual rainfall exceeds 600 mm, like in the semi-arid tropical regions (*de Tafur, El-Sharkawy & Calle, 1997*), and in region with over 1,000 mm of annual rainfall, like in the sub-humid and humid tropical regions (*Pellet & El-Sharkawy, 1997*). Excessive rainfall, however, can result in flooding and yield losses (*IPCC, 2021*).

Cassava production in Africa is also significantly constrained by cassava brown streak disease (CBSD), caused mainly by the cassava brown streak virus and its Ugandan variant, which are both members of family *Potyviridae* and genus *Ipomovirus* (*Winter et al., 2010*). Interestingly, to date and to the best of our knowledge, CBSD is confined to Africa only. For over 70 years, the disease was distributed only along the East African coast (*Tomlinson et al., 2018*). Earlier reports indicated that altitudes greater than 1,000 m above sea level (masl) were unfavourable for the survival of the disease (*Nichols, 1950*). Cold temperatures were also believed to enhance symptom severity and lead to the death of infected plants through die-back, starting from the shoot tip (*Jeremiah et al., 2015*). However, from 2004, CBSD outbreaks were reported in regions of Uganda and other parts of East and Central Africa

above 1,000 masl (*Alicai et al., 2007*). The CBSD outbreaks above the previous attitudinal ceiling (*i.e.,* <1,000 masl) were associated with an elevated vector population, whitefly *Bemisia tabaci,* in these regions. Whiteflies have become better adapted to conditions prevalent in cassava-growing zones of the Great Lakes region (800 to 1,500 masl) (*Jeremiah et al., 2015*).

Cassava brown streak viruses can be introduced into cassava plantations in two ways: through immigrating infectious whiteflies and/or through the use of infected cuttings (*Donnelly, Sikazwe & Gilligan, 2020*). The former is a more important secondary spread within the farm (*i.e.,* for short-distance movement) whereas the latter is more important for both the short and long-distance spread of the viruses. Indeed, the use or sharing of infectious planting materials for establishing a new field, especially for vertically transmitted plant diseases, plays a significant role in the dispersal of pathogens. This is especially important when the extent of using infected materials is commonly practiced at places where the abundance of infected whiteflies is low. This also may introduce pathogens to other distant countries where the viruses are currently absent. Also, the network over which the farms are connected will then facilitate pathogen dispersal, and needs to be considered when attempting to control or contain the disease (*McQuaid, Gilligan & Van den Bosch, 2017*). Cassava brown streak viruses are transmitted by whiteflies semi-persistently; they can be acquired from infected plants and transmitted to healthy plants within 5–10 minutes (*Donnelly, Sikazwe & Gilligan, 2020*; *Maruthi et al., 2016*). The probability of virus transmission, thus, increases with the number of whiteflies. This explains why the areas with high virus pressures are typically associated with the "superabundance" of whiteflies (*Donnelly, Sikazwe & Gilligan, 2020*; *Maruthi et al., 2016*).

Whiteflies (family, Aleyrodidae) are a complex of morphologically indistinguishable cryptic species with distinct geographical, biological, and genetic variations (*Mugerwa et al., 2012*). *Bemisia tabaci* species, attacks crops, weeds, and ornamental plants at all growth stages, feeding on the phloem of host plants. The pest causes direct and indirect damage to cassava through feeding and honeydew secretion, respectively (*Campo, Hyman & Bellotti, 2011*). Direct damage occurs as *Bemisia tabaci* species suck sap from leaves and induces physiological disorders. During feeding, these insects secrete honeydew which prevents leaves from functioning efficiently and acts as a substrate for fungi (sooty moulds). Large populations of whiteflies develop early in the crop's leaf, reducing plant vigour and tuber size, and leading to plant stunting (*Omongo et al., 2012*). Nevertheless, the factors contributing to the current and future geographic distribution of *Bemisia tabaci* species, especially in the cassava-growing regions of Africa, remain understudied. The life cycle of a whitefly is highly dependent on temperature, precipitation, and the host plant. High temperatures lead to a faster development and an increased insect population density (*Katono et al., 2022*). On the other hand, rainfall negatively affects the whitefly population (*Naranjo & Ellsworth, 2005*). A notable increase in whitefly abundance has been recorded throughout the cassava-growing regions of East and Central Africa since the 1990s (*Colvin et al., 2004*; *Legg et al., 2011*; *Otim et al., 2006*), with the local density on cassava shoots also increased from a few adults to hundreds (*Legg et al., 2006*). Few studies currently focus on assessing the geographic distribution of whitefly pests in Africa (*Jarvis et al., 2012*).

In African cassava growing areas, five members of the *Bemisia tabaci* species complex have been identified based on mitochondrial cytochrome oxidase I (COI) sequences, including Saharan Africa 1 to 5 (SSA1-5). SSA1 is further divided into five sub-groups: SSA1 sub-group1 (SSA1-SG1), SSA1-SG2, SSA1-SG3, SSA1-SG4 and SSA1-SG5 (*Boykin & De Barro, 2014*; *Mugerwa et al., 2012*; *Mugerwa et al., 2021*). However, SSA1 and SSA2 are more prevalent whiteflies associated with the spread of cassava mosaic disease (CMD) and cassava brown streak disease (CBSD) in East Africa (*Casinga et al., 2022*; *Mugerwa et al., 2012*). In addition, SSA1-SG1 to SG3 is the most predominant mitotype in East and Central Africa, including the Democratic Republic of Congo (DRC) and its neighbouring countries (Burundi, Central African Republic, Rwanda, Tanzania, and Uganda) (*Casinga et al., 2022*; *Mugerwa et al., 2012*) except South Sudan, which has SSA2 as the dominant mitotype on cassava. However, the Global Biodiversity Information Facility (GBIF) do not distinguish between members of the *Bemisia tabaci* species complex. This is partly because the identification of members of the *Bemisia tabaci* species complex is complicated. The technology for the identification of the *Bemisia tabaci* species complex is relatively new (*Boykin & De Barro, 2014*). As such, the data used in this study represents all members *Bemisia tabaci* species complex in Africa.

Species distribution models (SDM) are a powerful tool for understanding factors that drive species distribution in a geographic region (*Mudereri et al., 2021*). They offer a means to study the projected impact of climate change on the geographic distribution of plants, pathogens, and pests. The model generates categories of climate suitability and matches these to geographical regions (*Ramos et al., 2019*). A few SDMs have been developed for predicting the geographic range of cassava and whiteflies (*Campo, Hyman & Bellotti, 2011*; *Jarvis et al., 2012*). In particular, *Campo, Hyman & Bellotti (2011)* used the maximum entropy (Maxent) algorithm in the SDM to evaluate the global distribution of four key biotic constraints: whiteflies, cassava green mites, cassava mosaic disease (CMD) and CBSD; they identified parts of Brazil, the Americas, Africa's rift valley, the southern tip of India, and much of Southeast Asia as potential hotspots for cassava pests and disease outbreaks. Using the EcoCrop model and 24 Global Circulation Models (GCMs) to project into the future climate, *Jarvis et al. (2012)* examined the impact of climate change on cassava in Africa and reported $-3.7\%$ to $+17.5\%$ changes in climate suitability across the continent. How these anticipated changes will affect the suitability of cassava, whitefly, and CBSD, as well as their interactions, has not been quantified.

Here, we use historical climate data (from https://worldclim.org/) that represent different rainfall and temperature regimes to explain the current distribution of cassava, *Bemisia tabaci* species, and CBSD in Africa. To explore the potential future distributions, we use climate projections from the phase-6 "BCC-CSM2-MR" GCM of the Coupled Model Intercomparison Project (CMIP 6), together with two scenarios of its Shared Socio-economic Pathway (SSP126 and SSP585) for year 2050 (average for 2041 to 2060) and year 2070 (average for 2061 to 2080). These climate scenarios show that annual temperatures and precipitations are expected to shift considerably in the future decades (*Almazroui et al., 2020*). For model fitting, we collate occurrence records of each species from the Global Biodiversity Information Facility (GBIF) and published literature (cf. Materials and

Methods section). Our dataset does not allow us to distinguish between the two viral species of the disease (the cassava brown streak virus and its Ugandan variant), and we thus refer to them collectively as CBSD. We use an ensemble of four SDM methods for model fitting, including boosted regression trees (brt), maximum entropy (Maxent), generalised additive model (gam), and multivariate adaptive regression splines (mars). The results highlight important factors responsible for the current distributions of cassava, whiteflies, and CBSD. Suitable habitats for cassava cultivation and those at risk of invasion by whiteflies and CBSD are mapped, to guide policy and adaptation planning for governments, private organisations, and farmers.

## MATERIALS AND METHODS

### Occurrence data collection

We obtained occurrence data in Africa from the Global Biodiversity Information Facility (GBIF), including 1,422 records for cassava (accessed 14 June 2021, http://dx.doi.org/10.15468/dl.98jqqy), and 1,169 records of *Bemisia tabaci* species (accessed 17 June 2021, http://dx.doi.org/10.15468/dl.755ck5). For quality assurance, we excluded records without geolocation and only kept unique records to avoid duplication. The occurrence records of CBSD were integrated from published literature (*Alicai et al., 2019*), GBIF, and country-wide surveys from the Cassava Diagnostic Project (CDP) in East Africa (accessed 21 November 2021 *via* AgShare.Today). The CDP involved seven countries, namely Tanzania, Kenya, Uganda, Rwanda, Mozambique, Malawi, and Zambia, following a published sampling protocol (*Alicai et al., 2019*). Through the Tanzania Agricultural Research Institute (TARI), we only accessed data from Tanzania and Uganda, which included CDP surveys from the National Crops Resources Research Institute (NaCRRI, Uganda) between 2004 and 2017. The data collection protocol are described in *Alicai et al. (2019)*. In these surveys, cassava fields were randomly selected along motorable roads at intervals of 7–10 km and up to 20 km, depending on the density of cassava plants. A farmer was identified and asked for consent to survey the field at each location. Fields with crops between 3–6 months after planting were selected for the surveys, as CBSD foliar symptoms become apparent at this stage and before leaf shedding. Field location coordinates were collected using handheld GPS devices. In each field, 30 plants of the predominant variety were surveyed along two diagonal transects in an X-shape, with 15 representative plants each transect. Each sampled plant was scored for severity of foliar and stem symptoms on a 1–5 scale, where 1 indicates no visible symptoms and 5 corresponds to pronounced/extensive vein yellowing, chlorotic blotches on leaves or severe lesions, streaks on stems, or defoliation and die-back (*Alicai et al., 2019*).

To assess the impact of *Bemisia tabaci* species lumping, we divided whitefly occurrence records into datasets based on the geographic distribution of members of *Bemisia tabaci* species. The first dataset includes occurrence records from East Africa (Uganda, Tanzania, Kenya, Rwanda, Burundi, Malawi, Madagascar, Mozambique, and South Sudan). This dataset represents members of Sub-Saharan Africa 1 (SSA1) that are prevalent and widespread in this region (*Mugerwa et al., 2012*). The second dataset included occurrence

records for Central and West Africa (Cameroon, Nigeria, Benin, Central Republic, DR Congo, Egypt Zambia and South Africa). This dataset represents members of Sub-Saharan Africa 2-5 (SSA2-5) and other *Bemisia tabaci* species that are widespread in this region. Then, the two datasets were compared with the full dataset from the African continent.

## Environmental variables

The extent of our current study is Africa, the second-largest continent in both area and human population, stretching 30.4 million km$^2$ in land area with six distinct climate zones: the equatorial, humid tropical, tropical, semi-desert (Sahalian), Mediterranean, and desert (*Beck et al., 2018*). North Africa has an arid desert climate characterised by high temperatures and little precipitation. Equatorial West and Central Africa have a monsoon climate characterised by high temperatures, soaring humidity, and heavy seasonal rains (*Chemura, Schauberger & Gornott, 2020*). East Africa is characterised by dry and rainy seasons, while the southern part of Africa is generally more temperate (*Nicholson, 2017*). For the current climate (*Hijmans et al., 2005*), we used the 19 interpolated bioclimatic variables of 1970–2000 from the WorldClim database (https://worldclim.org/) at 10-min (~340 km$^2$) resolution (Table 1). We also included other biologically relevant predictors, including cassava harvested area (CHA), elevation (elev) and seven measures of soil quality (sq1-7) (Table 1). The CHA is an essential determinant of disease presence, and we used a standardised, high-quality, representative cassava map from *Szyniszewska (2020)*, accessible *via* Figshare repository (10.6084/m9.figshare.9745118). Elevation can influence the occurrence and dispersal of *Bemisia tabaci* species by altering precipitation, temperature, vegetation, including crops, and the angle, direction, and intensity of the solar radiation (*Mudereri et al., 2021*; *Ramos et al., 2018*). Elevation data was downloaded from the digital elevation model (DEM) of the shuttle radar topographic mission (https://srtm.csi.cgiar.org/), available at approximately 90 m pixel size with a vertical error of less than 16 m. Although cassava is a resilient crop, adaptable to diverse and poor soils, the quality of soils substantially improves the crop's productivity. We used seven key soil qualities that are important for crop production: nutrient availability, nutrient retention capacity, rooting conditions, oxygen availability to roots, excess salts, toxicities, and workability, denoted as sq1 to sq7, respectively, from the FAO soils portal (https://www.fao.org/soils-portal) at a 30 arc-sec (about 1 km$^2$) resolution (*FAO, 2000*).

To assess the potential future distributions of cassava, its pests *Bemisia tabaci* species, and the CBSD outbreaks, we used the ''BCC-CSM2-MR'' climate model developed by the Beijing Climate Center of the China Meteorological Administration (*Wu et al., 2019*). The model was chosen based on a comprehensive assessment of the predictive capacity of eight GCMs (Figs. AC1-8, AD1-8, & AE1-8) on the three target species (cassava, whitefly, and CBSD). Our assessment revealed no marked differences in their ability to predict the future distribution of the target species. Although this model is part of 49 climate models included in the 2021 IPCC sixth assessment report (AR6), only eight models with complete coverage were available from the WorldClim 2.1 database (https://worldclim.org/) (Table 2). We used predictions from this model under two SSPs (SSP1-2.6 and SSP5-8.5) for year 2050 (mid-century average for 2041-2060) and year 2070 (near late twenty-first-century average

**Table 1 Environmental variables.** A list of environmental data layers used for characterisation of cassava, whitefly and cassava brown streak disease (CBSD) in Africa and the percent contribution of each variable.

| Code | Variable description | Unit | Relative importance | | |
|------|----------------------|------|---------|----------|------|
| | | | Cassava | Whitefly | CBSD |
| Bio01 | Annual Mean Temperature | °C | – | – | – |
| Bio02 | Mean Diurnal Range | °C | 1.4% | 2.2% | 6.9% |
| Bio03 | Isothermality | °C | 31.6% | 3.8% | 2.3% |
| Bio04 | Temperature Seasonality | °C | 20.3% | – | – |
| Bio05 | Max Temperature of Warmest Month | °C | – | – | – |
| Bio06 | Min Temperature of Coldest Month | °C | – | – | – |
| Bio07 | Temperature Annual Range | °C | – | – | – |
| Bio08 | Mean Temperature of Wettest Quarter | °C | 6.5% | – | 2.6% |
| Bio09 | Mean Temperature of Driest Quarter | °C | – | – | – |
| Bio010 | Mean Temperature of Warmest Quarter | °C | – | – | – |
| Bio011 | Mean Temperature of Coldest Quarter | °C | – | – | 4.6% |
| Bio012 | Annual Precipitation | mm | – | 6.3% | 9.7% |
| Bio013 | Precipitation of Wettest Month | mm | 13.0% | 13.5% | 2.5% |
| Bio014 | Precipitation of Driest Month | mm | 5.6% | 13.6% | 9.8% |
| Bio015 | Precipitation Seasonality | mm | 1.9% | – | – |
| Bio016 | Precipitation of Wettest Quarter | mm | – | – | – |
| Bio017 | Precipitation of Driest Quarter | mm | – | – | – |
| Bio018 | Precipitation of Warmest Quarter | mm | 0.7% | 6.6% | 4.4% |
| Bio019 | Precipitation of Coldest Quarter | mm | 2.0% | 9.3% | 0.9% |
| Sq1 | Nutrient availability | – | 3.9% | – | – |
| Sq2 | Nutrient retention capacity | – | 1.4% | – | – |
| Sq3 | Rooting conditions | – | 0.6% | – | – |
| Sq4 | Oxygen availability to roots | – | 0.4% | – | – |
| Sq5 | Excess salts | – | 2.4% | – | – |
| Sq6 | Toxicity | – | – | – | – |
| Sq7 | Workability | – | 0.1% | – | – |
| Elev | Elevation; Ground height above sea level | m | – | 23.6% | – |
| CHA | Cassava harvested area | km$^2$ | – | 8.9% | 14.6% |

for 2061-2080). The SSP1-2.6 scenario is part of the "sustainability" SSP1 socio-economic family, representing the best case where the best policies are implemented. Conversely, the SSP5—8.5 scenario represents the worst-case scenario with high fossil fuel consumption throughout the 21st century and without climate mitigation policies (*Meinshausen et al., 2020*), resulting in global warming ranging from a low of 3.1 °C to a high of 5.1 °C by 2100.

## Species distribution modelling

Before building the SDM, out of 28 predictors we selected a subset according to the variance inflation factor (VIF) to avoid high level of collinearity that increases the uncertainty in model parameters and decreases the efficiency and power of model predictions (*Naimi &*

**Table 2 Modelling methods.** The bold values indicate that they are selected for the ensemble model. A single star in data type column indicates that model uses presence-background (pb*) data whereas double stars show that the model uses presence-only (p**) data.

| Method | Model class/description | Data type | AUC |
|---|---|---|---|
| RF | machine learning; random forest | pb* | 0.994 |
| **BRT** | **machine learning; boosted decision trees** | **pb*** | **0.985** |
| **MAXENT** | **machine learning; maximum entropy** | **pb*** | **0.983** |
| **SVM** | **machine learning; support vector machine** | **pb*** | **0.976** |
| CART | machine learning; classification and regression trees | pb* | 0.966 |
| BIOCLIM | profile/envelope model | p ** | 0.773 |
| MAHAL | profile/envelope model; mahalanobis | p ** | 0.993 |
| DOMAIN | profile/envelope model | p ** | 0.942 |
| GAM | regression; generalised additive model | pb* | 0.981 |
| GLM | regression; generalised linear model | pb* | 0.969 |
| **MARS** | **regression; multivariate adaptive regression splines** | **pb*** | **0.976** |
| GLMNET | regression; Lasso and Elastic-Net Regularized Generalized Linear Models | pb* | 0.941 |

*Araújo, 2016*; *De Marco & Nóbrega, 2018*). We calculated the VIFs of all predictors using the "usdm" R-package and excluded the one with the greatest VIF sequentially till all remaining predictors have VIFs <10 (*Naimi & Araújo, 2016*). As a result, 15 predictors were selected for cassava, 9 predictors for whitefly and 10 predictors for CBSD.

We developed the SDM using the "sdm" package (*Naimi & Araújo, 2016*) in the R platform (R version 4.0.5; *R Core Team, 2021*). The sdm package provides an object-oriented, reproducible, and extensible platform, capable of handling an ensemble of models, and we evaluated a total of twelve algorithms for their ability to predict the potential distribution of the three species of interest (Table 3). The models are classified based on the nature of data used *i.e.,* presence-only *vs* presence-background data (*Elith et al., 2006*). As the species occurrence data consists of presence-only records, we randomly added 1,000 pseudo-absences throughout the study area (*Thuiller, Georges & Engler, 2013*). Variable importance is a metric used to determine the contribution of predictor variables in explaining the species distribution (*Naimi & Araújo, 2016*). In the "sdm" package, this is handled using "getVarImp" function. To assess model performance, we used the conventional metrics of the area under the receiver operator characteristic (ROC) curve (AUC), which measures the discriminatory ability of each model (*Jiménez-Valverde, 2012* and *Nahm, 2022*). AUC is a threshold-independent performance measure that reflects the probability of a randomly chosen presence site ranking above a background site. The AUC value is usually divided into five levels (*Guan et al., 2021*): 0–0.6 (fail), 0.6–0.7 (poor), 0.7–0.8 (fair), 0.8–0.9 (good), and 0.9–1 (excellent). In this study, the models with AUC outputs in the range of 0.97 <AUC<1 were selected for further analysis. A fivefold cross-validation technique with five repetitions was used to assess the model performance (*Thuiller, Georges & Engler, 2013*). Specifically, models were calibrated on a random sample of 80% of the occurrence data and evaluated on the remaining 20% (*Ramos et al., 2018*; *Ramos et al., 2019*).

**Table 3  Suitability habitats.** Current and future suitable habitats for (a) cassava, (b) whitefly, and (c) cassava brown streak disease (CBSD) as predicted by version 2 of Beijing Climate Center Climate System Model (BCC-CSM2-MR). The suitability scores are defined as: (0–0.2) Unsuitable, (0.2–0.4) Low suitability, (0.4–0.6) Moderate suitability, (0.6–0.8) Suitable and (0.8–1.0) Very suitable.

| Scenario | Time | Unsuitable (x10⁶km²) | Low (x10⁶km²) | Moderate (x10⁶km²) | Suitable (x10⁶km²) | Very suitable (x10⁶km²) |
|---|---|---|---|---|---|---|
| A. Cassava | | | | | | |
| Current | 1970-2000 | 13.6(45%) | 3.4 (11%) | 5.9 (20%) | 4.8 (16%) | 2.1 (7%) |
| SSP 126 | 2050s | 12.0(40%) | 5.3 (18%) | 2.3 (8%) | 3.0 (10%) | 7.1 (24%) |
|  | 2070s | 12.6(42%) | 4.7 (16%) | 2.4 (8%) | 3.0 (10%) | 7.2 (24%) |
| SSP 585 | 2050s | 12.9(43%) | 4.5 (15%) | 2.5 (8%) | 3.1 (10%) | 6.8 (23%) |
|  | 2070s | 13.3(45%) | 4.0 (13%) | 2.4 (8%) | 2.9 (10%) | 7.3 (24%) |
| B. Whitefly | | | | | | |
| Current | 1970-2000 | 18.7(63%) | 3.1(11%) | 2.5 (8%) | 1.9 (6%) | 3.6 (12%) |
| SSP 126 | 2050s | 16.7(56%) | 1.8 (6%) | 3.2 (11%) | 4.3 (14%) | 3.9 (12%) |
|  | 2070s | 19.6(57%) | 1.9 (6%) | 2.7 (11%) | 5.0 (14%) | 2.4 (12%) |
| SSP 585 | 2050s | 19.7(58%) | 2.0 (7%) | 2.7 (11%) | 4.9 (14%) | 2.5 (10%) |
|  | 2070s | 19.3(59%) | 2.2 (7%) | 2.9 (12%) | 5.1 (14%) | 2.4 (8%) |
| C. CBSD | | | | | | |
| Current | 1970-2000 | 19.8(66%) | 4.4(15%) | 2.9 (10%) | 1.9 (6%) | 8.0 (3%) |
| SSP 126 | 2050s | 13.4(45%) | 10.3(35%) | 2.9 (10%) | 1.9 (6%) | 8.0 (3%) |
|  | 2070s | 13.8(46%) | 10.1(34%) | 4.6 (8%) | 2.2 (4%) | 1.5 (8%) |
| SSP 585 | 2050s | 14.6(49%) | 9.8 (33%) | 4.8 (8%) | 2.0 (4%) | 1.3 (6%) |
|  | 2070s | 15.1(50%) | 10.2(34%) | 4.9 (8%) | 1.8 (3%) | 1.0 (5%) |

# RESULTS

## Model selection and performance

Of the 28 predictors, different sets were selected after eliminating the effect of multicollinearity. For cassava, 15 predictors (Bio02–04, 08, 13–15, 18–19; Sq1–5, 7) were selected for modelling potential distributions. Nine predictors were selected for modelling whitefly distributions (Bio02–03, 12–14, 18–19; CHA, and elev), while 10 were selected for modelling CBSD (Bio02–03; 08, 11–14, 18–19; and CHA). Of the 12 SDMs evaluated (Table 3), nine of the models had an AUC score above 90%, however, only four models (brt, maxent, gam, and mars) performed the best (AUC>97%) and were selected for further analyses. These four top models produced, on average, an AUC of 98.0% for cassava, 98.3% for *Bemisia tabaci* species and 99.5% for CBSD. The performance of the ensemble models on two groups of species (SSA1 and SSA2-5) is provided in Appendix F. We excluded the other models from the ensemble partly because we considered them either overfitting (AUCs closest to 100%, including random forest and Mahal.dismo) or their AUCs less than 97% (including, GLMNET and Bioclim). The average of these four top models was used for mapping the current and future suitability habitats of targeted species.

## Predictors explaining distributions of cassava, whiteflies, and CBSD

For cassava, isothermality (Bio03, relative importance: 31.6%), temperature seasonality (Bio04, 20.3%), precipitation of wettest month (Bio13, 13.0%), and mean temperature of

wettest quarter (Bio08, 6.5%) were the top predictors explaining the observed distribution (Table 1). For whiteflies, elevation (23.6%), precipitation of the driest month (Bio14, 13.6%), precipitation of the wettest month (Bio13, 13.5%) and precipitation of the coldest quarter (Bio19, 9.3%) strongly impacted the current distribution (Table 1). In contrast, cassava harvested area (CHA, 14.6%), precipitation of driest month (Bio14, 9.6%), annual precipitation (Bio12, 9.7%) and mean diurnal range (Bio02, 6.9%) were largely responsible for the distribution of CBSD outbreaks (Table 1).

The probability of occurrence responded to different environmental predictors in a nonlinear way (Fig. 1). Specifically, for cassava (Fig. 1A), the occurrence probability tends to increase with isothermality (Bio03), and precipitation of the wettest month (Bio13), but decline with mean temperature of wettest quarter (Bio08). For whiteflies (Fig. 1B), the occurrence probability tends to increase with precipitation of wettest month (Bio13) and precipitation of driest month (Bio14), while decline with cassava harvested area (CHA). In contrast, cassava harvested area (CHA) and precipitation of driest month (Bio14) showed a positive relationship with the occurrence probability of CBSD (Fig. 1C), while the occurrence probability showed a negative relationship with annual precipitation (Bio12) and mean diurnal range (Bio02).

## Current and potential future suitable habitats for cassava in Africa

Cross-validation showed that the predicted range of suitable habitats based on 80% training records is consistent with the observed range from 20% testing records of cassava (Fig. 2A), with 54.6% (16.2 million km$^2$) of the continent suitable for cassava production. These suitable habitats (*i.e.,* suitability above 0.2) were predicted to locate predominantly in Sub-Saharan Africa. The regions with moderate suitability (*i.e.,* 0.4–0.6), corresponding to 10.7 million km$^2$, were also found mainly in Sub-Saharan Africa. Although our data do not contain records from countries such as South Sudan, Sudan, Somalia, Botswana, and Zimbabwe, suitable habitats were predicted in these places. For habitats that are classified as 'very suitable' (suitability above 0.8), a total area of approximately 2.1 million km$^2$ was identified, predominantly in Coastal Guinea, Sierra Leone, Ivory Coast, Ghana, Togo, Benin, central to southern Nigeria, Cameroon, south-west Central African Republic, southern-western parts of South Sudan, Rwanda and the northern parts of Zambia (Figs. 2A–2B). Under the two SSP scenarios (SSP1.2-6 and SSP5.8-5) a potential increase of 59.6% and 56.6% in suitable habitats for cassava was projected for the mid-term (2050), predominantly in Sub-Saharan Africa (Figs. 2C–2D & Table 3). This trend was found to decline slightly towards 2070 (an increase of 57.8% and 55.3% compared to the current distribution). Specifically, highly suitable area (suitability above 0.8) for cassava production is expected to expand, from the East Coast of South Africa to Mozambique, and northern Madagascar (Appendix C; Fig. C1-8; for contracted areas, see Table 4).

## Current and potential future regions at risk of whitefly invasion

Approximately 37.3% of Africa's land area is predicted suitable for whiteflies, with a total of approximately 3.6 million km$^2$ highly suitable (>0.8), covering Uganda, southern Ethiopia, Rwanda, some parts of Tanzania, the Northern province of Zambia, southern DR Congo,

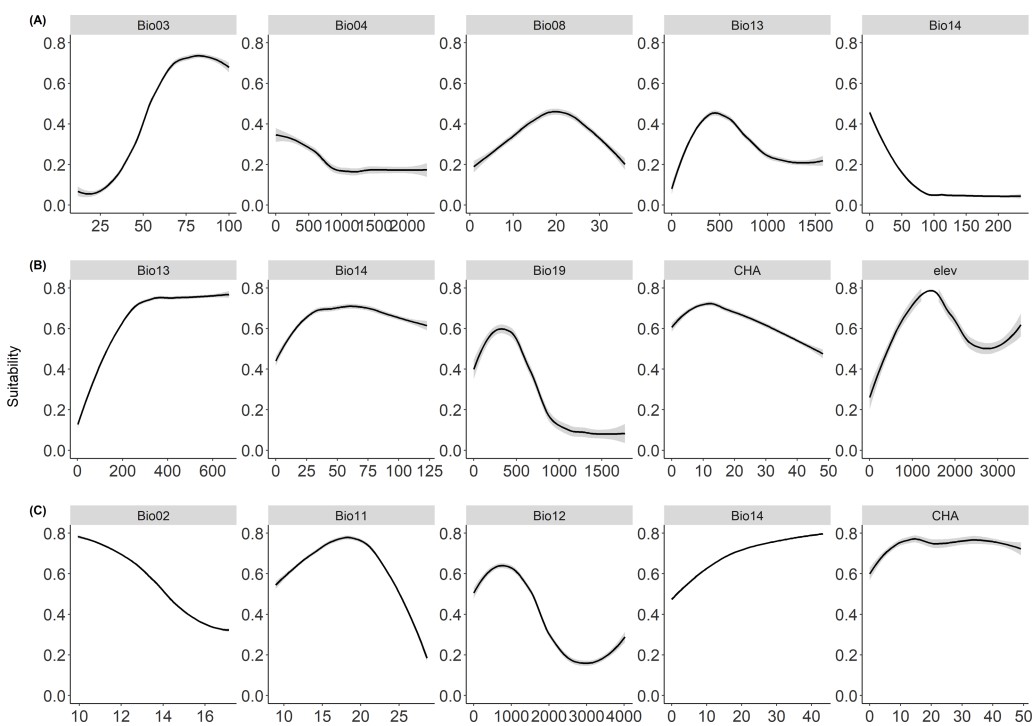

**Figure 1** **Response curves.** Figure showing the response curves for (A) cassava, (B) whitefly, and (C) cassava brown streak disease showing the probability of occurrences with response to predictor variables. The full names of the variables are present in Table 1.

central Cameroon, and some parts of Nigeria (Figs. 3A & 3B). For both SSP scenarios (SSP1-2.6 and SSP5-8.5), the ensemble model from the four selected SDM algorithms predicted an increase of highly suitable areas by 44.1% and 44.8%, respectively, by 2050 (Figs. 3C–3D), although the expansion was predicted to decline towards 2070 (an increase by 43.4% and 40.9% compared to the current distribution) (Appendix D; Fig. D1-8). The region with the highest future potential of whitefly infestation includes primarily East African countries, specifically South Sudan, Ethiopia, Zambia, Malawi, Angola, Central Cameroon, south-west Nigeria, Ghana, and Ivory Coast (cf. Table 5 for a summary of current and potential future suitable regions in Africa).

## Current and potential future hotspots for cassava brown streak disease

The area currently susceptible to CBSD covers approximately 33.7% of Africa's land area (10.2 million km$^2$; Fig. 4A). The countries most vulnerable to this disease include the east coast and Lake zones of Tanzania, Uganda, and Southeast DRC (Table 5). Although no occurrence records were available in western Africa, the ensemble models highlighted suitable conditions in this region, specifically in Ivory Coast, Ghana, Nigeria (highest), and Cameroon, and East Africa emerged as a hotspot for CBSD outbreaks (Appendix E; Fig. E1-8). Under two SSP scenarios (SSP1-2.6 and 5-8.5) CBSD is expected to expand its range to 55% (16.6 million km$^2$) and 56.6% (10.2 million km$^2$) of Africa's land area,

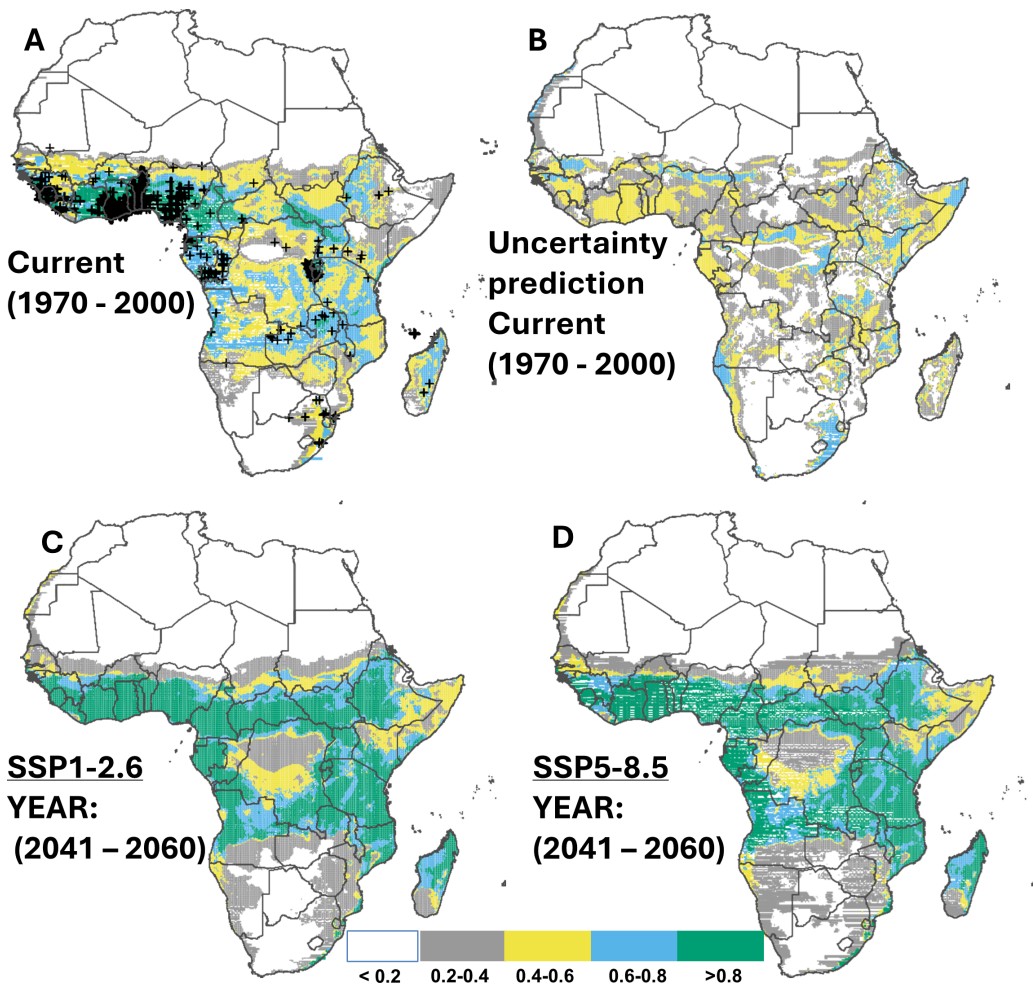

**Figure 2 Current and future suitable habitats for cassava in Africa.** Plots showing (A) the predicted distribution for cassava under the current climate (1970–2000), (B) prediction uncertainty for cassava under the current climate, (C) predicted future suitable habitats for cassava under the SSP1-2.6 scenario and (D) predicted future suitable habitats for cassava under the SSP5-8.5 scenario using version 2 of the Beijing Climate Center Climate System Model (BCC-CSM2-MR) for the year 2050. Black crosses indicate GBIF records. White colour represents unsuitable, grey colour represents low suitability, yellow colour represents moderate suitability, sky blue represents suitable, bluish green colour represents very suitable. The corresponding long-term future prediction (2070s) of suitable habitats for cassava under the SSP1-2.6 and SSP5-8.5 are presented in Appendix C Fig. C1-8.

respectively, by 2050 (Figs. 4B–4C), and this predicted range remains largely unchanged towards 2070 (Appendix E; Fig. E1-8), with countries most susceptible including those where the disease is yet to be reported, including Ivory Coast, Ghana, Benin, Nigeria, and Cameroon, all located in West Africa which is key for current cassava production. Other suitable areas were also found in southern Africa, specifically southwestern DRC, northern Angola, northern Zambia, and the eastern shores of Mozambique (Table 5).

**Table 4 Change in suitable habitats.** Predicted future change in suitable habitats for (a) cassava, (b) whitefly, and (c) cassava brown streak disease (CBSD) using shared social economic pathways (SSPs); SSP1-2.6 and SSP5-8.5 for 2041-2060 and 2061-2080. The suitability scores are; ($< -0.2$) Contraction, ($>$ +0.2) Expansion and ($-0.2$–+0.2) Unchanged.

| Scenario | Time | Contraction (x$10^6$km$^2$) | Expansion (x$10^6$km$^2$) | Unchanged (x$10^6$km$^2$) |
|---|---|---|---|---|
| A. Cassava | | | | |
| SSP 126 | 2050s | 4.8 (16.0%) | 14.8 (49.2%) | 10.4 (34.8%) |
| | 2070s | 4.8 (16.2%) | 14.6 (49.0%) | 10.4 (34.8%) |
| SSP 585 | 2050s | 5.5 (18.3%) | 14.7 (49.1%) | 9.7 (32.6%) |
| | 2070s | 5.4 (18.0%) | 15.4 (51.5%) | 9.1 (30.5%) |
| B. Whitefly | | | | |
| SSP 126 | 2050s | 7.4 (24.7%) | 2.4 (7.9%) | 20.1 (67.4%) |
| | 2070s | 7.6 (25.3%) | 2.3 (7.6%) | 20.0 (67.1%) |
| SSP 585 | 2050s | 8.1 (27.2%) | 2.4 (8.0%) | 19.3 (64.8%) |
| | 2070s | 8.5 (28.6%) | 2.6 (8.7%) | 18.7 (62.7%) |
| C. CBSD | | | | |
| SSP 126 | 2050s | 10.8 (36.3%) | 12.8 (42.8%) | 6.2 (20.9%) |
| | 2070s | 10.9 (36.6%) | 12.5 (42.0%) | 6.4 (21.4%) |
| SSP 585 | 2050s | 11.3 (38.0%) | 12.0 (40.4%) | 6.5 (21.6%) |
| | 2070s | 12.0 (40.0%) | 11.7 (39.3%) | 6.2 (20.7%) |

## Impact of *Bemisia tabaci* species lumping in predicting the current and future suitable habitats

Sub-Saharan Africa 1 (SSA1) *Bemisia tabaci* species are currently distributed and widespread in East Africa, and are likely to remain restricted to this region in future (Appendix F; Figs. F1A & F1D). On the other hand, the other *Bemisia tabaci* species, Sub-Saharan Africa 2-5 (SSA2-5) are currently distributed across Central and some countries in West Africa. With increasing warming in future, the suitable habitats of SSA2-5 species are likely to shrink to Ivory Coast, Ghana, Togo and Benin (Appendix F; Figs. F1B & F1E). Compared with maps produced using occurrence records from two geographical regions, the SDM model built using full *Bemisia tabaci* species records yielded consistent predictions for the current and future scenarios (Appendix F; Figs. F1C & F1F).

## DISCUSSION

Using the most recent climate scenarios from CMIP6 for the mid-term (2041–2060) and long-term (2061–2080) in Africa, our study provides a comprehensive overview of suitable habitats for cassava, as well as one of its economically important diseases (CBSD) and vector agent (*Bemisia tabaci* species). The results indicate that under current conditions, Sub-Saharan Africa remains a hotspot for cassava, its associated virus/disease, and the vector agent (*Szyniszewska, 2020*; *Tomlinson et al., 2018*). These results are congruent with the fact that cassava is already widely cultivated in Sub-Saharan Africa, especially western Africa, with Nigeria being the highest producer, producing more than 6 million tons per annum (*FAO, 2013*; *FAO, 2000*). This is in contrast with the identified high suitability areas

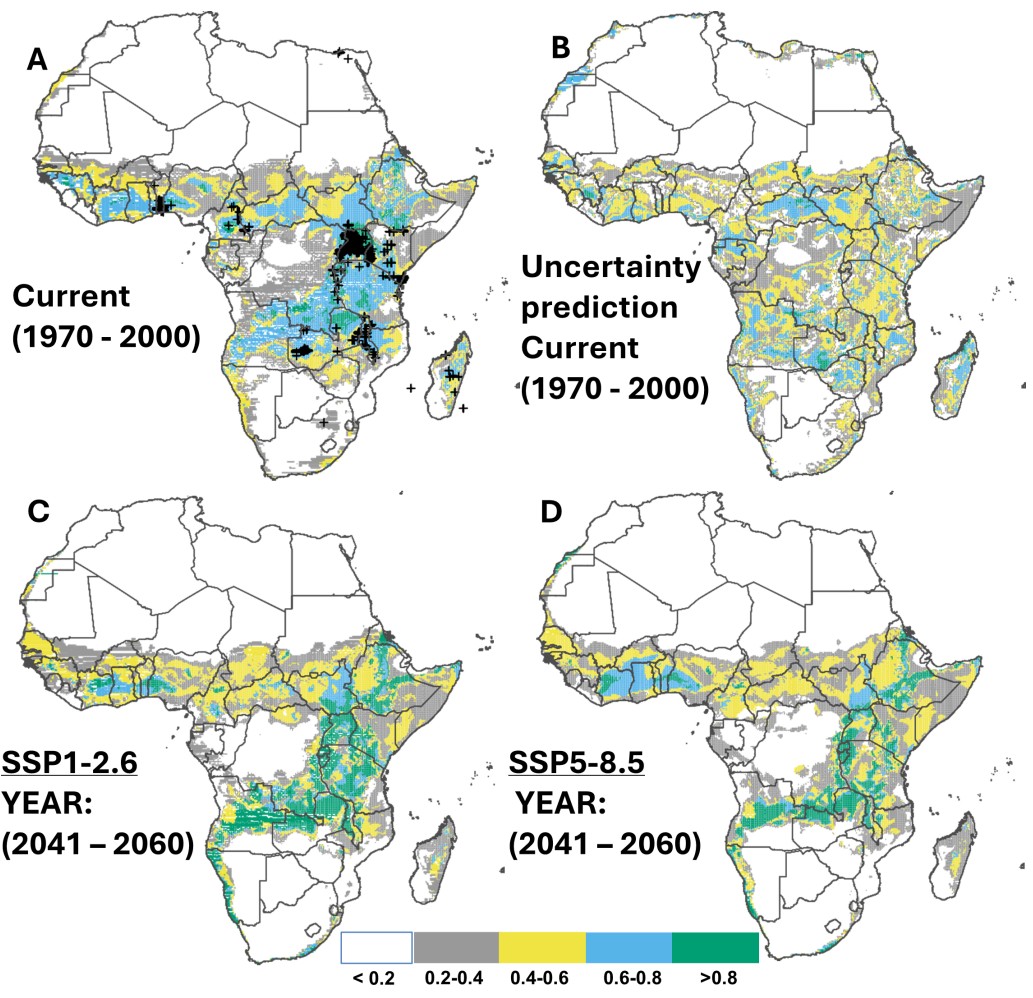

**Figure 3  Current and future suitable habitats for whitefly (*Bemisia tabaci*) in Africa.** Plots showing (A) the predicted distribution for whitefly under the current climate (1970–2000), (B) prediction uncertainty for whitefly under the current climate, (C) predicted future suitable habitats for whitefly under the SSP1-2.6 scenario and (D) predicted future suitable habitats for whitefly under the SSP5-8.5 scenario using version 2 of the Beijing Climate Center Climate System Model (BCC-CSM2-MR) for the year 2050. Black crosses indicate GBIF records. White colour represents unsuitable, grey colour represents low suitability, yellow colour represents moderate suitability, sky blue represents suitable, bluish green colour represents very suitable. The corresponding long-term future prediction (2070s) of suitable habitats for whitefly under the SSP1-2.6 and SSP5-8.5 are presented in Appendix D, Fig. D1-8.

that are largely restricted to Eastern Africa (Tanzania and Uganda). This partially reflects the history that CBSD originated from the Coastlines of Eastern Africa before spreading to Middle Africa despite eradication measures to delimit its spread (*Tomlinson et al., 2018*). This westward spread of the virus remains of great concern. To date, the limited whitefly population in other cassava growing countries like Nigeria has not been accompanied by reports of the disease (*Nwezeobi et al., 2020*). The SDM ensemble was successful, in terms of model performance (*Guan et al., 2021*), in reconstructing these patterns for species of interest; it identified new potential suitable habitats for cassava planting, whitefly invasion,

**Table 5   Classification of African countries by suitability score.** Current and potential future regions with suitable and highly suitable areas for cassava, whitefly and cassava brown streak disease in Africa.

| Class | Description | Countries under current climate | Countries under future climate |
|---|---|---|---|
| **Cassava** | | | |
| 0.8–1.0 | Very suitable | Sierra Leone, Ivory coast, Ghana, Togo, Bennin, Nigeria, Cameron, West Central Republic, and Rwanda | Guinea, Sierra Leone, Ivory Coast, Ghana, Togo, Bennin, Nigeria, Cameron, Gabon, Republic of Congo, Central Republic, South Sudan, Ethiopia, Kenya, Uganda, Tanzania, Rwanda, Burundi, Mozambique, Zambia, Angola and Madagascar |
| 0.6–0.8 | Suitable | Mali, Chad, DR Congo, Angola, Kenya, Tanzania, Ethiopia, Zambia, Malawi, Zimbabwe and Madagascar | Sudan, Somalia, Chad |
| Whitefly | | | |
| 0.8–1.0 | Very suitable | Uganda, North Zambia, Central Tanzania, Rwanda, and Central Cameron | Ivory coast, Angola, South DR Congo, Zambia, Malawi, West Namibia, Tanzania, Rwanda, Burundi, Kenya, Uganda, and Ethiopia |
| 0.6–0.8 | Suitable | Guinea, Ivory coast, Ghana, Togo, Bennin, Nigeria, Cameron, Central Republic, South Sudan, Kenya, DR Congo, Angola, West Zambia, Malawi, and Madagascar | Ghana, Central Republic, South Sudan, Madagascar, and Mozambique |
| CBSD | | | |
| 0.8–1.0 | Very suitable | Uganda, Tanzania, South DR Congo | Uganda, Tanzania, Kenya, Rwanda, Burundi, South DR Congo, Angola, Ethiopia, Cameron, Nigeria, Ghana, and Ivory coast |
| 0.6–0.8 | Suitable | Kenya, Ethiopia, Mozambique, Zambia, Central Republic, Nigeria, Ghana, and Ivory coast | Mozambique, Somalia, Central Republic and Bennin |

and CBSD prevalence in future scenarios, which are informative for local and regional decision making relating to food security.

## Factors of habitat suitability

We further found that different predictors are differentially driving the distribution of all three species of interest under current conditions. Only Bio 14 (precipitation of the driest month) was, to some extent, driving the distribution of all three species. This was more important for whitefly than for cassava. For cassava, we found temperature fluctuations and seasonality to have the strongest influence. Being a drought-tolerant plant, cassava can survive a wider range of environmental conditions, hence fluctuations, than insects (*El-Sharkawy, 2004*). The reproductive rate of whitefly is also said to decline with extreme temperatures (*Aregbesola et al., 2020*). As the carrier of the virus, the implication is that virus incidences will likely be reduced under unfavourable climate conditions, especially high temperatures (*El-Sharkawy, 2004*). Such high temperatures and precipitation events reported in Western Africa in this current term (*Almazroui et al., 2020*; *Campo, Hyman & Bellotti, 2011*) can explain the low prevalence of the vector and virus in this and other regions depicted by the current distribution.

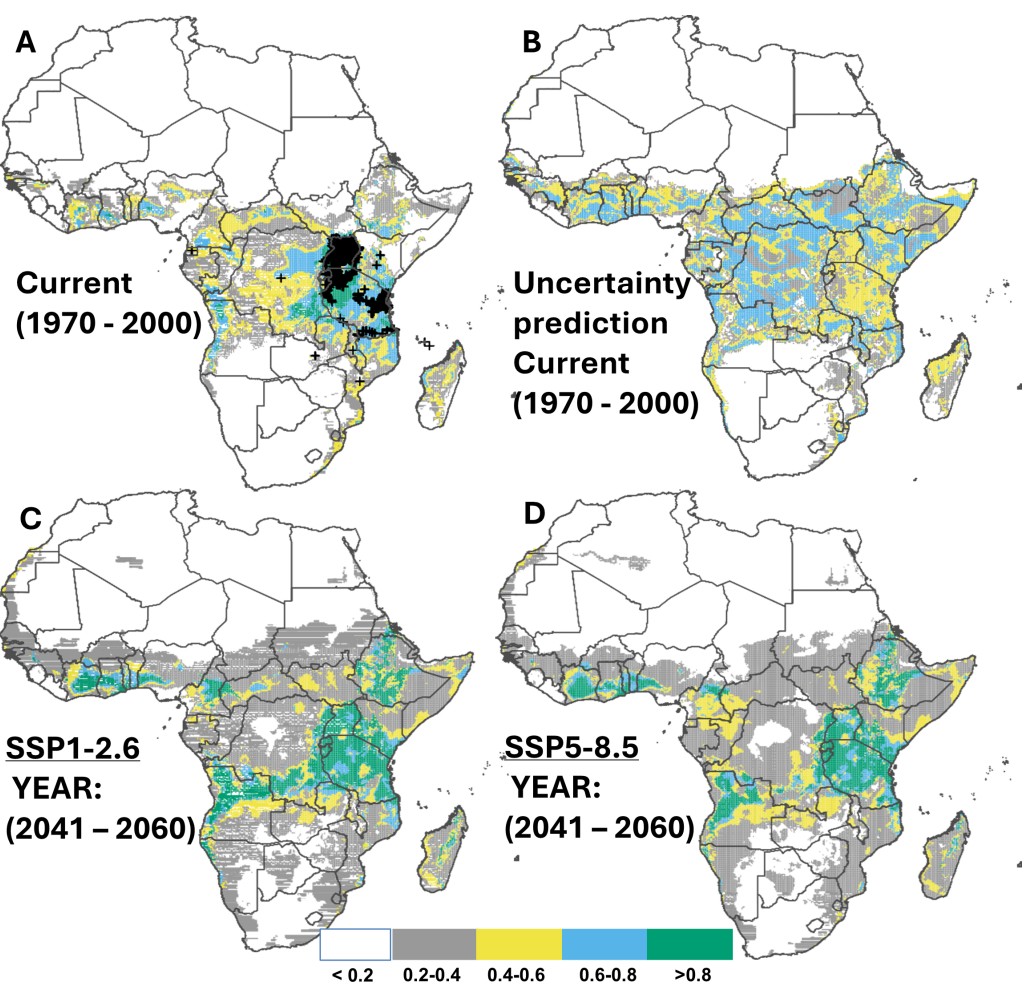

**Figure 4 Current and future suitable habitat for cassava brown streak disease (CBSD) in Africa.** Plots showing (A) the predicted distribution for CBSD under the current climate (1970–2000), (C) prediction uncertainty for CBSD under the current climate (C) predicted future suitable habitats for CBSD under the SSP1-2.6 scenario and (D) predicted future suitable habitats for CBSD under the SSP5-8.5 scenario using version 2 of the Beijing Climate Center Climate System Model (BCC-CSM2-MR) for the year 2050. Black crosses indicate GBIF records. White colour represents unsuitable, grey colour represents low suitability, yellow colour represents moderate suitability, sky blue represents suitable, bluish green colour represents very suitable. The corresponding long-term future prediction (2070s) of suitable habitats for CBSD under the SSP1-2.6 and SSP5-8.5 are presented in Appendix E Fig. E1-8.

As found in our study and consistent with the literature, elevation had the strongest influence on the distribution of whitefly but not cassava and CBSD. *Jeremiah et al. (2015)* found that whiteflies were more abundant in mid-altitude areas, *i.e.,* 1,000–1,500 masl. However, areas with higher altitudes (>1,500 masl; Fig. 1B) are less suitable for whitefly, given their low temperatures (*MacFadyen et al., 2018*). Nevertheless, the presence of cassava fields will, to an extent, influence the occurrence of whitefly and spread of the disease (*Donnelly & Gilligan, 2020*). This was evident in our study by the contribution of cassava harvested area to the current distribution of both species. This contribution was at

a lower magnitude for whitefly but was the highest driver of CBSD. It again amplifies the fact that under favourable environmental conditions, other aspects like infested cassava planting materials can continue to maintain the disease even in the absence of whitefly if the diseased cuttings are not adequately disposed of *McQuaid, Gilligan & Van den Bosch (2017)*. This also explains the high vulnerability to the disease in East Africa.

**Future scenarios of cassava planting**

With all models and climate scenarios, Sub-Saharan Africa especially will become increasingly suitable for cultivation of cassava. This includes even small parts east of southern Africa. This is positive news in view of the importance and extensive use of cassava as a food security crop and for biofuel production (*Nweke, 2004*). This observation is consistent with other studies, which reported that suitable habitats for Cassava would increase across Africa (*Jarvis et al., 2012*). However, the above authors also reported that cassava suitability habitats would decrease in Central and West Africa. This contrasts with our findings which depict expansions across a greater proportion of Saharan Africa. A few reasons can be attributed to this discrepancy. *Jarvis et al. (2012)* used future climate information derived from an older phase of the CMIP project (CMIP 3, SRES-A1B emissions scenario). According to the climate modelling community, the CMIP3/CMIP5 over- or under projects climate changes in certain areas (*Brown et al., 2013*). Also, it was observed that while CMIP5 models did not capture the observed increase in global mean surface temperature between 1998 and 2013, the historical CMIP6 simulations are able to reproduce these observed recent temperature increases (*Bock et al., 2020*). More generally, CMIP6 has a higher ability to reproduce observed large-scale mean surface temperature and precipitation patterns compared to their CMIP3 and CMIP5 counterparts (*Bock et al., 2020*). Additionally, CMIP6 factors social-economic and policy aspects (*Tebaldi et al., 2021*; *Zelinka et al., 2020*). These authors also only used an EcoCrop model, while we used a more robust ensemble of four models.

The increased cassava suitability habitats reported in this paper also highlights Cassava's ability to withstand climate change relative to other crops like maize, millet, sorghum, banana, and beans (*Jarvis et al., 2012*; *Kotir, 2011*). Nevertheless, this does not imply that climate mitigation actions should be ignored in favour of increasing the suitable habitats for Cassava, as our results do not speak to how final agronomic outputs will be affected. Crops are not only affected by temperature, but also generally exhibit different responses to increased $CO_2$ concentration. For Cassava, it is postulated that temperature increases could negatively affect biomass and yields due to poor assimilation (*Amuji, 2021*). Climate reports indicate that under these shared socio-economic pathway scenarios, temperature levels will already be well above 1.5 °C and 2 °C above pre-industrial periods (*Osima et al., 2018*). These increases are firmly cautioned against in the Paris Agreement on climate change (*UNFCCC, 2016*).

Increases in suitable habitats for cassava also correlate positively with slight increases in whitefly density and slighter increases in CBSD density, as seen in outputs from our future projections. This supports the likelihood of westward spread of the disease, especially along the coastal parts of Western Africa (*Bundi et al., 2022*). However, the tropical rainforest

and monsoon conditions found in the DRC, especially central DRC, and zones along the Sahel belt may continue to have a positive impact in reducing the vulnerability of the region to both whitefly and hence CBSD. This is consistent with what was postulated by *Campo, Hyman & Bellotti (2011)*—that whitefly distribution is limited by extreme dryness along the Sahel belt and extreme and prolonged wetness in parts of central Africa. More so, even suitable areas for Cassava would be much reduced in central DRC and along the Sahel hence further limiting disease outbreaks. Adequate pest mitigation measures at international borders, coupled with good farm management strategies, would therefore largely favour a reduced vulnerability to whitefly and CBSD in these regions. The conclusions from this study corroborate a modelling study (using a different approach) looking at historical climate change in the East Africa region (*Kriticos et al., 2020*). The maps for the current distribution of *Bemisia tabaci* species show large areas suitable for whitefly species but without GBIF records. One of the reasons is attributed to the robustness of our model prediction. The ensemble modelling approach used could extrapolate suitable habitats for whiteflies beyond their current geographical range. This was also true for the prediction of cassava brown streak disease. Indeed, CBSD can be introduced into new areas *via* planting materials. Using cassava harvested area (CHA) as a covariate we found suitable areas in West Africa where the disease is currently absent (Appendix B; Fig. B4). This agrees with other modelling studies evaluating the role of mixed modes of infection in the spread of CBSD. They found that infected planting material through trade was the key long-distance pathogen dispersal mechanism (*McQuaid, Gilligan & Van den Bosch, 2017*).

The current distribution of the entire African species complex seems to be a composite of the current distributions of SSA1 and SSA2-5. However, there is a possibility that the southern strip in the distribution from Zambia through Angola to the coast of Namibia for the species complex has been overestimated. This overestimation might be attributed to the southern records of SSA2-5 being geographically connected with those of SSA1, resulting in an inflated southeast strip of high suitability in the distribution of the species complex.

It is worth noting that many SDMs target taxonomic units above the species level, and statistically, there is no limitation on using SDMs beyond the species level (*Stas et al., 2020*). While this approach is commonly used when dealing with a functional guild, it has been questioned when dealing with cryptic species. Overall, lumping species of *Bemisia tabaci* could lead to an overestimation of habitat suitability in areas with records of mixed lineages. This overestimation could be problematic as these mixed lineages may respond to climate change independently. Therefore, there is a need for more precise species identification in these areas.

One of the limitations of this study is the lack of species-specific data for whitefly, *Bemisia tabaci*, and cassava brown streak virus on GBIF. This led to lumping all *Bemisia tabaci* species and species causing CBSD together. However, *Bemisia tabaci* is now recognized as a pest species complex of over 25 species (*Tay et al., 2022*). In East Africa, two distinct species of *Bemisia tabaci* have been identified, including sub-Saharan Africa 1 (SSA1) and one from South West Indian Ocean (SWIO) islands (*Mugerwa et al., 2012*). The availability of these data in GBIF would improve the prediction of species distribution models. Given the

generic approach used in this study (*Campo, Hyman & Bellotti, 2011*; *Muderi et al., 2021*; *Ramos et al., 2018*), we believe that the resulting maps provide a good approximation of the true distribution of the species complex, but can be refined in the future for species-specific mapping.

Although Africa is one of the largest cassava producers, occurrence records in online repositories do not reflect what is known about cassava production (*Szyniszewska et al., 2021*). It is reported that less than 4% of the total records published on GBIF pertain to Africa. To bridge the gap of data deficit, efforts should be increased to encourage data capturing and sharing in order to promote more evidence-based studies. Future studies should also focus on how well these cultivars can perform under these climate change conditions under controlled greenhouse conditions to understand if the increased suitability habitats will be commensurate with yield increases.

## CONCLUSION

We have shown in this study using recent climate scenarios that climate change will continue to render the African continent vulnerable to whitefly and the cassava brown streak disease in the mid-century (*i.e.,* by the year 2041-2060). This vulnerability will spread to the coastal parts of Western Africa as cassava-suitable habitats increase. While this increase in cassava suitable habitats sounds encouraging, the associated risk of disease outbreaks needs to be monitored and potentially mitigated. It, therefore, remains imperative that as the growing areas for cassava are exploited in the future, more stringent measures should be applied to delimit the occurrence and spread of whitefly and the cassava diseases caused by the brown streak virus.

Some tentative measures include, among others: (i) stricter national and international border controls and quarantine measures on the movement of plant materials as this is one of the most common methods through which diseased materials have been introduced to the continent and several regions (*McQuaid, Gilligan & Van den Bosch, 2017*); (ii) in already heavily infested countries, infested planting and farm materials should be incinerated or properly sterilised, while clean materials produced for instance through tissue culture techniques, should be made readily available to farmers more frequently; (iii) to reduce the populations of whiteflies, bioagents could be identified and continuously introduced in a controlled manner; (iv) increase the use of improved cassava varieties that are tolerant/resistant to both whitefly and CBSD; (v) build the capacity of extension officers and farmers on the on-farm identification of CBSD symptoms and whitefly irradiation measures; (vi) increase all efforts along the climate change mitigation pipeline to reduce warming and greenhouse gas emissions at both local and international levels. Farm practices can be modified to facilitate these measures. For example, intercropping cassava with maize in the Ivory Coast proved to reduce the incidence of cassava mosaic disease (*Lapidot et al., 2014*). Phyto-sanitation practices can be enforced whereby infected plants are uprooted and removed from the farm to prevent them from acting as a source of inoculum (*Legg et al., 2017*).

Overall, to prevent disease outbreaks and associated costs from yield loss, CBSD-resistant/tolerant varieties should be deployed in areas that are highly suitable for cassava

production but where there is a low probability of disease occurrence. There are several cassava varieties with varying levels of resistance (*Sheat et al., 2019*). However, the most sustainable strategy to control CBSD is using resistant varieties and supplying disease-free planting materials to cassava farmers. Breeding of resistant varieties is achieved using new breeding technologies that require a short breeding period, unlike conventional strategies that require 6 to 8 years before a new variety can be developed (*Bizimana et al., 2024*). These technologies include marker-assisted selection, genomic selection, transgenesis, genome editing and others that have been developed and applied to cassava. Although in some cases resistant varieties support large whitefly populations, using resistant varieties mitigate/prevent the damage associated with whitefly abundance. To mitigate actual costs from yield loss, these varieties can also be deployed in suitable areas of cassava production but currently experience high disease outbreaks. Countries that are highly suitable for cassava production include most of West Africa (Table 5), where there is currently a low chance of disease incidence. These varieties should be incorporated into farmer-preferred varieties to increase the chances of acceptability. In addition, resources to check for infected cuttings should be invested at the borders of DR Congo with Congo and the Central African Republic. These are the two potential entry borders to West Africa, given that CBSD is already reported to cause substantial yield loss in DR Congo.

### Funding

This work was supported by the African Institute for Mathematical Sciences, with financial support from the Government of Canada, provided through Global Affairs Canada and the International Development Research Centre. David Richardson received support from Mobility 2020 project no. CZ.02.2.69/0.0/0.0/18_053/0017850 (Ministry of Education, Youth and Sports of the Czech Republic) and long-term research development project RVO 67985939 (Czech Academy of Sciences). Pietro Landi and Cang Hui are supported through the research program 'Advancing Biodiversity Informatics and Ecological Modelling' at the National Institute for Theoretical and Computational Sciences (NITheCS). Cang Hui is also supported by the National Research Foundation (NRF grant 89967). The funders had no role in study design, data collection and analysis, decision to publish, or preparation of the manuscript.

### Grant Disclosures

The following grant information was disclosed by the authors:
The African Institute for Mathematical Sciences.
The Government of Canada, provided through Global Affairs Canada and the International Development Research Centre.
Mobility 2020 project no. CZ.02.2.69/0.0/0.0/18_053/0017850 (Ministry of Education, Youth and Sports of the Czech Republic).
Czech Academy of Sciences: RVO 67985939.

'Advancing Biodiversity Informatics and Ecological Modelling' at the National Institute for Theoretical and Computational Sciences (NITheCS).
The National Research Foundation: 89967.

## Competing Interests

The authors declare there are no competing interests.

## Author Contributions

- Geofrey Sikazwe conceived and designed the experiments, performed the experiments, analyzed the data, prepared figures and/or tables, authored or reviewed drafts of the article, and approved the final draft.
- Rosita Endah epse Yocgo conceived and designed the experiments, analyzed the data, prepared figures and/or tables, authored or reviewed drafts of the article, and approved the final draft.
- Pietro Landi conceived and designed the experiments, authored or reviewed drafts of the article, and approved the final draft.
- David M. Richardson conceived and designed the experiments, authored or reviewed drafts of the article, and approved the final draft.
- Cang Hui conceived and designed the experiments, authored or reviewed drafts of the article, and approved the final draft.

## Data Availability

The source maps are available at Figshare: Sikazwe, Geofrey (2023). CassavaMapAfrica. figshare. Figure. https://doi.org/10.6084/m9.figshare.22726121.v3

Sikazwe, Geofrey (2023). BtabaciMapAfrica. figshare. Figure. https://doi.org/10.6084/m9.figshare.19518835.v1

Sikazwe, Geofrey (2023). CBSDMapAfrica. figshare. Figure. https://doi.org/10.6084/m9.figshare.22722607.v1.

The R code is available at Figshare: Sikazwe, Geofrey; Hui, Cang; Landi, Pietro (2023). R code. figshare. Software. https://doi.org/10.6084/m9.figshare.24428041.v1.

## Supplemental Information

Supplemental information for this article can be found online at http://dx.doi.org/10.7717/peerj.17386#supplemental-information.

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
