# Peer review of "Current and future scenarios of suitability and expansion of cassava brown streak disease, Bemisia tabaci species complex, and cassava planting in Africa"

_PeerJ, doi:10.7717/peerj.17386_

## Round 0.1 · original submission · Minor Revisions

Dear Dr. Sikazwe,

After this first review round, both reviewers positively assessed your manuscript. Still, Reviewer #1 raised some interesting points that should be considered before the manuscript is formally accepted for publication in PeerJ. In addition to the issues raised by R#1, I would like to point out the need to include some arguments in your text covering the ins and outs of lumping different species into a single taxonomic unit when modeling species distributions. As soon as all of these issues are appropriately covered, I believe your text will be accepted for publication in PeerJ.

Sincerely,
Daniel Silva

Reviewer 1 ·

Basic reporting

This is a straight forward manuscript which details the potential distribution of cassava, whitefly pests, and a cassava disease (CBSD) across Africa under future climate change scenarios. This is an area that requires more research and the authors provide modelling outputs that could be used for multiple applications.

The fact that CBSD can also be transmitted by the use of infected cuttings needs to be explained in the introduction (and this may be a more important mode of transmission in low whitefly density regions). I would also explain that CBSD viruses are semi-persistent and do not enter the circulatory system of whitefly pests. They basically sit on the whitefly mouthparts for a few weeks so it is not an easy mechanism to transmit virus from from plant to another. It may be that we need high densities of whitefly to see CBSD transmission (given the low infection rate).

Experimental design

The major limitation of this SDM approach is introduced by the decision to lump all the whitefly species together, and lumping the two different viruses that cause CBSD together. Bemisia tabaci is now recognized as a pest species complex of over 25 species (see Tay WT, Court LN, Macfadyen S, et al (2022) A high-throughput amplicon sequencing approach for population-wide species diversity and composition survey. Molecular Ecology Resources 22:1706–1724. https://doi.org/10.1111/1755-0998.13576
). The GBIF records do not distinguish these species, but attempts are being made to resolve the species distributions across East Africa. It is becoming clear that some species (like SSA1) are well adapted to cassava, whilst other species may not be. I appreciate the authors can't yet get detailed presence/absence data for individual whitefly species and viruses but I question how meaningful the resulting maps and predictions can be given this generic approach. I suggest the authors address this challenge clearly in the discussion and provide a thoughtful analysis of how this might impact conclusions.

Validity of the findings

The overall conclusions from the modelling seems valid, and corroborate a modelling study (using a different approach) looking at historical climate change in the East Africa region (see: Kriticos DJ, Darnell RE, Yonow T, et al (2020) Improving climate suitability for Bemisia tabaci in East Africa is correlated with increased prevalence of whiteflies and cassava diseases. Scientific Reports 10:. https://doi.org/10.1038/s41598-020-79149-6).

I would like the authors to add the GBIF records to the current day distribution maps (figures), so we can see where there is currently suitable conditions but yet not record of whitefly, cassava or CBSD. This would help give the reader confidence in the models.

Elevation being an important factor related to whitefly is not a very novel finding, and the exact mechanisms underlying this pattern can't be untangled. But it does give confidence that the model is providing a useful reflection of true suitability.

Additional comments

Adding some more complex economic scenarios to this modelling, or adding details about where CBSD resistant varieties should be deployed to limit spread (ahead of the disease arriving) would make this modelling exercise more useful for decision-making. Saying exactly where resources should be invested to check for infected cuttings at borders (given you won't be able to cover all border areas) would also be a useful application of these models. Such applications would make the models and resulting maps more useful and less descriptive. The authors could add these ideas into the discussion rather than just focussing on methods to manage the pests themselves.

Reviewer 2 ·

Basic reporting

Relatively good writing observed

Experimental design

The four species distribution models (boosted regression trees, maximum entropy, generalized additive model and multivariate adaptive regression splines as well as environmental covariates are relevant for the studies. Well suited. Good choices

Validity of the findings

The findings appear valid and scientifically sound

Additional comments

No comment

---

## Round 0.2 · Major Revisions

Dear Dr. Sikazwe,

After this first review round, although one of the reviewers recommended acceptance, the other one rejected your manuscript. I agree with the main issues raised by the review. Still, I wanted to provide a second chance to deal with such issues before rejecting the manuscript.

Please note that you need to address more properly the issues related to the lumping of different species during the SDM procedures and what this can cause to your general results. For instance, I would like to see other examples of studies that lumped different species as a single operational unit in SDMs. Additionally, I would like to see a more profound discussion of what this procedure can result in terms of SDM results in general.

Since you are dealing with SDMs, you need to take extreme care in curating the occurrences of your modeled species to avoid producing unreliable results. This caution is extremely important to be considered in general.

Sincerely,
Daniel SIlva

Reviewer 1 ·

Basic reporting

The authors have made limited changes to the MS to accommodate some of the suggestions I have made but have not really incorporated the changes into their interpretation of the model outputs or their discussion of findings.

Specifically:
1. They acknowledge that B. tabaci is a pest species complex, but then refer to it as a singular species in the title, abstract, introduction etc. It is only in the conclusion that they disclose this critical fact. This is a major flaw of the MS that is technically incorrect. It is relatively easy to fix, but it isn't a minor change.

2. They acknowledge in the introduction that CBSD can be transferred via infected cuttings, but again don't link this back to what it means for the model outputs.

The authors have added the GBIF records on top of the model predictions, and there are large areas that are suitable for whitefly species but without GBIF records. This needs to be discussed.

Experimental design

See flaws identified above.

Validity of the findings

I'm not sure the recommendation to release CBSD tolerant or resistant varieties is valid as I'm not sure there are many resistant varieties to release (see: https://www.ncbi.nlm.nih.gov/pmc/articles/PMC6523400/). My understanding is that there are some varieties that are a bit more tolerant than others but no high levels of resistance. Furthermore some of the CMD resistant varieties that have been released are very good at supporting whitefly populations. So variety release is perhaps something you want to explain in more detail.

Additional comments

Please don't put limitations of your study in the conclusions section.

Reviewer 2 ·

Basic reporting

no comment

Experimental design

very good. well elaborated

Validity of the findings

valid

Additional comments

no comment

---

## Round 0.3 · accepted · Accept

Dear Dr. Sikazwe,

I am pleased to accpet your manuscript for publication in PeerJ.

Congratulations,
Daniel Silva

Reviewer 1 ·

Basic reporting

Looks like they have really addressed the species complex issue. The manuscript looks much better and is more logical. My other comments have also been addressed.

Experimental design

no comment

Validity of the findings

no comment